# Power Allocation Algorithm for an Energy-Harvesting Wireless Transmission System Considering Energy Losses

**Su Zhao [1,2,\*], Gang Huang [1,2] and Qi Zhu [2]**

[1]   The National Mobile Communications Research Laboratory, Southeast University, Nanjing 210003, China; 13815863971@163.com
[2]   Jiangsu Key Lab of Wireless Communications, Nanjing University of Posts and Telecommunications, Nanjing 210003, China; zhuqi@njupt.edu.cn
\*   Correspondence: zhaos@njupt.edu.cn; Tel.: +86-189-5189-6599

**Abstract:** For an energy-harvesting wireless transmission system, considering that a transmitter which can harvest energy from nature has two kinds of extra energy consumption, circuit consumption and storage losses, the optimization models are set up in this paper for the purpose of maximizing the average throughput of the system within a certain period of time for both a time-invariant channel and time-varying channel. Convex optimization methods such as the Lagrange multiplier method and the KKT (Karush–Kuhn–Tucker) condition are used to solve the optimization problem; then, an optimal offline power allocation algorithm which has a three-threshold structure is proposed. In the three-threshold algorithm, two thresholds can be achieved by using a linear search method while the third threshold is calculated according to the channel state information and energy losses; then, the offline power allocation is based on the three thresholds and energy arrivals. Furthermore, inspired by the optimal offline algorithm, a low-complexity online algorithm with adaptive thresholds is derived. Finally, the simulation results show that the offline power allocation algorithms proposed in this paper are better than other algorithms, the performance of the online algorithm proposed is close to the offline one, and these algorithms can help improve the average throughput of the system.

**Keywords:** energy harvesting; circuit consumption; storage losses; power allocation

---

## 1. Introduction

With the development of wireless communication technology, environmental protection and energy consumption issues have attracted widespread attention. The technology of green development and energy saving has become the mainstream technology of wireless communication development [1,2]. Energy harvesting is a method of resource regeneration in wireless communication, which can realize the efficient use of renewable resources. In a scenario that the system has energy-harvesting transmitters, energy harvesters can harvest energy from the natural environment (such as wind energy, solar energy, tidal energy, etc.). Since the amount of harvested energy is limited and has large randomness, it is important to decide how to allocate the energy to transmit data. Although some research has been undertaken into power allocation in energy harvesting systems, algorithms are usually proposed which fail to take the energy losses into account or only think about one kind of energy loss. It is worth researching power allocation when considering that several kinds of energy losses exist.

## 2. Related Work

There has been some research into wireless communication systems with energy-harvesting equipment. For a time-invariant channel, in [3], the optimal offline power allocation policies are proposed to minimize the transmission completion time under energy causality for two kinds of data arrival scenarios. In [4], the optimal power allocation policies are derived when the capacity of the battery is limited. In [5], for a sensor networks scenario, the problem of maximizing the throughput of the system is firstly studied, and then the problem of minimizing the average delay is solved. In addition, wireless transmission with a time-varying channel is also studied. In [6], directional water-filling algorithm is proposed to solve throughput maximization question and the proving of optimality is given. The authors next use stochastic dynamic programming to tackle the optimal online policy with causal channel information and propose a low-complexity near-optimal algorithm. In [7], structural results for the optimal energy allocation are obtained by using dynamic programming and optimization theory; then, the staircase water-filling algorithm is obtained based on the conventional water-filling algorithm when considering that the capacity of a battery has no limit. In [8], a geometric water-filling (GWF) algorithm is proposed in place of the conventional water-filling algorithm with a sum power constraint. Then, the recursively geometric water-filling (RGWF) algorithm is obtained by taking energy causality into account.

All the energy collected in [3–8] is used for transmission and other losses are ignored, which is too idealistic. In [9], the authors analyze the problem of optimizing system throughput in the case of imperfect batteries: the batteries will leak over time or the upper limit of the battery capacity decreases with time. In [10], for a time-invariant channel, taking the circuit consumption factor into account and considering both energy efficiency (EE) and spectrum efficiency (SE), the EE–SE two-phase algorithm is proposed to solve the problem of the optimization of throughput. In [11], circuit consumption is also considered for both time-invariant and time-varying channels, and the optimal solution is given to maximize the throughput. In [12], three questions are studied: throughput maximization, the minimization of energy consumption when data transmission is completed, and the minimization of data transmission completion time. Inspired by the offline optimal algorithm of the above problems, an online power allocation algorithm with low complexity is proposed. The losses of energy in the storage process are studied in [13], where the conditions of time-invariant and time-varying channels, infinite and limited battery capacity, single users and broadcast channels are discussed, respectively. Then, an optimal power allocation algorithm with a double-threshold structure is proposed. Besides the above-mentioned studies, there has also been some research into power allocation in other scenarios. For example, [14] develops distributed methods to efficiently use harvested energy in a sensor network while [15] considers power and server allocation in a multibeam satellite downlink.

Some of the above references do not consider the additional energy losses and some consider only the single energy losses, while there would be more than one kind of energy loss problem in the actual scene. This paper aims to study the scenario where not only circuit consumption generated in the process of transmission but also the energy losses in the storage process are considered. These two factors are considered as restrictive conditions in the optimization problem for time-invariant and time-varying channels, respectively. We discuss the offline optimal power allocation strategies to solve the throughput maximization problems and then propose an online power allocation policy with low complexity.

The contents are as follows: Section 3 shows the establishment of the system model; Section 4 solves the optimization problem for the time-invariant channel and proposes a power allocation algorithm with a three-threshold structure; Section 5 presents the optimal power allocation strategy in view of the time-varying channel; Section 6 puts forward the online power allocation algorithm with low complexity and adaptive thresholds; Section 7 gives the simulation results and analysis; and Sections 8 and 9 are a discussion and a summary of this paper, respectively.

## 3. System Model

The model of a wireless transmission system with an energy harvesting transmitter is shown in Figure 1. Supposing that the data transmission time has $N$ time slots, the length of each time slot is unit 1, the transmitter will harvest $E_i$ units of energy at the beginning of time slot $i$. The energy $E_i$ collected at each time slot is independent and obeys uniform distribution. The transmit power in time slot $i$ is $p_i$; when $p_i > 0$, the circuit consumption power is $a$ mW, and the total power consumed is

$$P_{all} = \left\{ \begin{array}{ll} p_i + a, & p_i > 0 \\ 0, & p_i = 0 \end{array} \right\} \tag{1}$$

In (1), $p_i$ is defined as

$$p_i = \frac{E_i - s_i + r_i - al_i}{l_i}, \quad i = 1, \ldots, N \tag{2}$$

where $l_i (0 \leq l_i \leq 1)$ is the actual transmission time, $s_i$ is the amount of energy stored in the battery, $r_i$ is the amount of energy extracted from the battery and $al_i$ is the consumption.

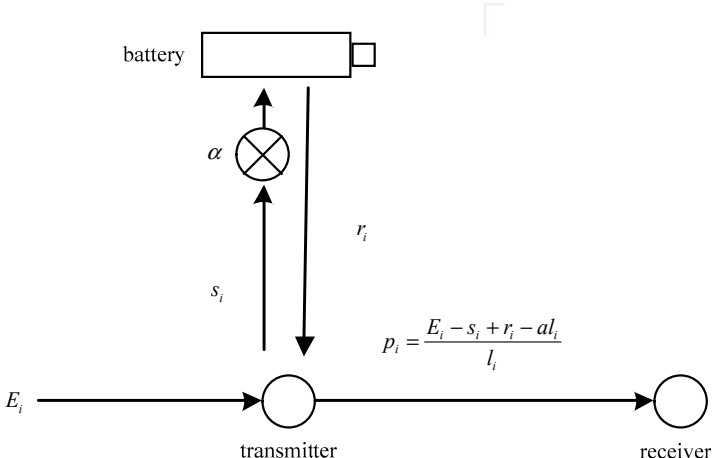

**Figure 1.** System model.

There are energy losses when the energy is stored in the battery, supposing that the storage efficiency is $\alpha$ $(0 \leq \alpha \leq 1)$. If the energy collected by the time slot is used up to transmit the data, there are no energy storage losses; if the collected energy is not used up at the current time slot, the remaining energy $s_i$ is stored in the battery, the amount of energy loss is $(1 - \alpha)s_i$, and only $\alpha s_i$ units of energy can be used for future. The amount of energy in the battery in time slot $i$ can be written as

$$B_i = \sum_{j}^{i} (\alpha s_j - r_j) \geq 0, \quad i = 1, \ldots, N \tag{3}$$

For the additive white Gaussian noise channel, the instantaneous transmission rate $R(p_i)$ is

$$R(p_i) = \frac{1}{2} \log(1 + h_i p_i) \tag{4}$$

In (4), $h_i$ is the channel fading coefficient in time slot $i$.

In this paper, the average throughput of the system is maximized by optimizing the transmit power of each time slot and the factors such as energy arrivals, channel states, circuit consumption and storage losses are considered comprehensively.

## 4. Optimal Offline Power Allocation for the Time-Invariant Channel

We first study the time-invariant channel scene where $h_i = h, i = 1, \ldots, N$ and the energy arrivals are $E_i, i = 1, \ldots N$. The instantaneous transmission rate is (4); then, the optimization problem P1 can be described as

$$\max_{\{s_i, r_i, l_i\}} \frac{1}{N} \sum_{i=1}^{N} l_i R\left(\frac{E_i - s_i + r_i - al_i}{l_i}\right) \tag{5}$$

$$\text{s.t.} B_i = \sum_{j=1}^{i} (\alpha s_j - r_j) \geq 0, \quad i = 1, \ldots, N \tag{6}$$

$$p_i = E_i - s_i + r_i - al_i \geq 0, i = 1, \ldots, N \tag{7}$$

$$s_i \geq 0, r_i \geq 0, 0 \leq l_i \leq 1, \ i = 1, \ldots, N \tag{8}$$

The condition in (6) means that the amount of energy in the battery is nonnegative, which reflects the energy causality. The condition in (7) means the power is nonnegative.

Observing this optimization problem, it can be found that $R(p)$ is concave in $p$ and all conditions are linear, so P1 is a convex optimization problem and the Lagrange multiplier method and KKT conditions can be used. The Lagrange function is

$$
\begin{aligned}
L = \sum_{i=1}^{N} \Big( \frac{1}{N} l_i R\left(\frac{E_i - s_i + r_i - al_i}{l_i}\right) + \quad & \lambda_i \left( \sum_{j=1}^{i} (\alpha s_j - r_j) \right) + \mu_i (E_i - s_i + r_i - al_i) \\
& v_i s_i +_i r_i + \theta_i l_i + \sigma_i (1 - l_i))
\end{aligned}
\tag{9}
$$

where $\lambda_i, \mu_i, v_{i,i}, \theta_i, \sigma_i, i = 1, \ldots N$ are nonnegative Lagrange multipliers for the restrictive conditions.

Taking the derivatives with respect to $s_i, r_i, l_i$, we can obtain

$$\frac{\partial L}{\partial s_i} = -\frac{h}{1 + hp_i^*} + \alpha \sum_{j=i}^{N} \lambda_j - \mu_i + v_i = 0 \quad i = 1, \ldots, N \tag{10}$$

$$\frac{\partial L}{\partial r_i} = \frac{h}{1 + hp_i^*} - \sum_{j=i}^{N} \lambda_j + \mu_i +_i = 0 \quad i = 1, \ldots, N \tag{11}$$

$$\frac{\partial L}{\partial l_i} = \frac{1}{2} \log(1 + hp_i^*) - \frac{h(p_i^* + a)}{2(1 + hp_i^*)} - a\mu_i + \theta_i - \sigma_i = 0, \quad i = 1, \ldots N \tag{12}$$

where $p_i^* = \frac{E_i - s_i^* + r_i^* - al_i^*}{l_i^*}$.

The remaining KKT conditions are

$$\lambda_i \sum_{j=1}^{i} (as_j^* - r_j^*) = 0, \quad i = 1, \ldots, N \tag{13}$$

$$\mu_i (E_i - s_i^* + r_i^* - al_i^*) = 0, \quad i = 1, \ldots, N \tag{14}$$

$$v_i s_i^* = 0, {}_i r_i^* = 0, \theta_i l_i^* = 0, \sigma(1 - l_i^*) = 0, i = 1, \ldots, N \tag{15}$$

Putting (10), (11) and (12) in order:

$$p_i^* = \frac{1}{\alpha \sum_{j=i}^{N} \lambda_j - \mu_i + v_i} - \frac{1}{h} = \frac{1}{\sum_{j=i}^{N} \lambda_j - \mu_i +_i} - \frac{1}{h}, \quad i = 1, \ldots, N \tag{16}$$

$$\frac{1}{2} \log(1 + hp_i^*) = \frac{h(p_i^* + a)}{2(1 + hp_i^*)} + a\mu_i - \theta_i + \sigma_i, \quad i = 1, \ldots, N \tag{17}$$

If $l_i{}^* = 0$, $p_i{}^* = 0$; If $0 < l_i{}^* \leq 1$, $p_i{}^* > 0$ and $\mu_i = \theta_i = 0$, (16) and (17) can be written as

$$p_i{}^* = \frac{1}{\alpha \sum_{j=i}^{N} \lambda_j + \nu_i} - \frac{1}{h} = \frac{1}{\sum_{j=i}^{N} \lambda_j + i} - \frac{1}{h}, i = 1, \ldots, N \tag{18}$$

$$\frac{1}{2} \log(1 + h p_i{}^*) = \frac{h(p_i{}^* + a)}{2(1 + h p_i{}^*)} + \sigma_i, i = 1, \ldots, N \tag{19}$$

We first analyze (18). Since the system exhibits storage losses, the optimal solution cannot be achieved by using a conventional water-filling algorithm or the DWF (Directional Water-Filling) algorithm in [6]. Reference [13] defines two thresholds according to (18):

$$p_{si} = \frac{1}{\alpha \sum_{j=i}^{N} \lambda_j} - \frac{1}{h}, i = 1, \ldots, N \tag{20}$$

$$p_{ri} = \frac{1}{\sum_{j=i}^{N} \lambda_j} - \frac{1}{h}, i = 1, \ldots, N \tag{21}$$

Obviously, $p_{si} > p_{ri}$, and the two thresholds have the following relationship:

$$\frac{1 + h p_{ri}}{1 + h p_{si}} = \alpha, i = 1, \ldots, N \tag{22}$$

In [13], the authors have analyzed $p_{ri} \leq p_i{}^* \leq p_{si}$, which means that the optimal power should be between two thresholds. If $p_i$ is greater than $p_{si}$, the part of the energy which exceeds $p_{si}$ should be stored in the battery, so $p_{si}$ is called the storage threshold; if $p_i$ is less than $p_{ri}$, the system needs to retrieve energy from the battery, so $p_{ri}$ is called the retrieval threshold. According to Lemma 3 in [13], the threshold should be monotonically increasing and its value changes only when the battery is empty. Combined with the analysis of [13] and the restrictions of this paper, the optimal power should satisfy the following conditions:

$$p_i{}^* = \min\left(p_{si}, \max\left(p_{ri}, [E_i / l_i{}^* - a]^+\right)\right) \tag{23}$$

$$s_i{}^* = [E_i / l_i{}^* - a - p_i{}^*]^+, \; r_i{}^* = [p_i{}^* - (E_i / l_i{}^* - a)]^+ \tag{24}$$

Then, we analyze (19) and firstly propose a new optimization problem, P2:

$$\max l_i \frac{1}{2} \log(1 + h p_i) \tag{25}$$

In (25), $l_i = \frac{E_i}{p_i + a}$, P2 aims to maximize the throughput in time slot $i$ with $E_i$ units of energy, and the length of time slot $i$ has no limit.

Taking the derivative of (25), we can obtain

$$\frac{1}{2} \log(1 + h p_i{}^*) = \frac{1}{2} \frac{h(p_i{}^* + a)}{(1 + h p_i{}^*)} \tag{26}$$

It is obvious that (26) is similar to (19), and the only difference is the parameter $\sigma_i$ which corresponds to the condition $l_i{}^* \leq 1$. Supposing that $p_o$ is the optimal solution of P2, if the circuit consumption $a$ and the channel fading coefficient $h$ are determined, $p_o$ is determined. It has been analyzed in [12] that the optimal power allocation for P2 adding the condition $l_i{}^* \leq 1$ should satisfy the following:

$$p_i{}^* = \max(p_o, E_i - a) \tag{27}$$

$$l_i{}^* = \frac{E_i}{p_i{}^* + a} \tag{28}$$

According to the above analysis, in our optimization problem, $p_i^*$ and $l_i^*$ should satisfy the following:

$$p_i^* = \max(p_o, E_i - s_i^* + r_i^* - a) \tag{29}$$

$$l_i^* = \frac{E_i - s_i^* + r_i^*}{p_i^* + a} \tag{30}$$

In addition, the last time slot should use up all the energy harvested and that remaining in the battery. Comprehensively considering the above analyses, finally, $p_i^*$ and $l_i^*$ should satisfy the following situations:

(1) When $pee > p_{si}$, then $p_i^* = p_o, i = 1, \ldots, N$. We first set $l_i^* = 1$, if $E_i - a > p_o$, $s_i^* = E_i - a - p_o, r_i^* = 0, l_i^* = 1$; if $p_{ri} \leq E_i - a \leq p_o$, $l_i^* = \frac{E_i}{p_o + a}, s_i^* = 0, r_i^* = 0$; if $E_i - a < p_{ri}$, $s_i^* = 0, r_i^* = \min(B_i, p_{ri} - (E_i - a)), l_i^* = \frac{E_i + r_i^*}{p_o + a}$.

(2) When $p_{ri} \leq p_o \leq p_{si}$, assume that $l_i^* = 1$. If $E_i - a > p_{si}$, $s_i^* = E_i - a - p_{si}, r_i^* = 0, l_i^* = 1$; if $p_{ri} \leq E_i - a \leq p_{si}$, $p_i^* = \max(p_o, E_i - a), s_i^* = 0, r_i^* = 0, l_i^* = \frac{E_i}{p_i^* + a}$; if $E_i - a < p_{ri}$, $p_i^* = p_o$, $s_i^* = 0, r_i^* = p_{ri} - (E_i - a), l_i^* = \frac{E_i + r_i^*}{p_i^* + a}$.

(3) When $p_o < p_{ri}$, we can obtain $p_i^*$ from (23) and $l_i^* = 1, i = 1, \ldots, N$.

No matter what the situation is, the power allocation in the last time slot should be

$$p_N^* = \max(p_o, (E_N + B_{N-1} - a)) \tag{31}$$

$$l_N^* = \frac{E_N + B_{N-1}}{p_o + p_N^*} \tag{32}$$

It can be found from the above discussion that power allocation mainly depends on $p_{si}, p_{ri}$ and $p_o$; thus, Algorithm 1 next proposed is called the three-threshold algorithm:

---

**Algorithm 1.** Optimal solution for the time-invariant channel

---

1) Calculate $p_o$, initialize:
$p_i = 0, r_i = 0, s_i = 0, l_i = 1, B_i = 0, i = 1, \ldots, N, ii = 1, k = 1$;
2) for $i = ii{:}1{:}N$ do
Linear search the largest $p_{s,k}$ and $p_{r,k}$ which can make $p_i, s_i, r_i, i = ii, \ldots, N$ feasible in (23), (24) and the condition $B_i \geq 0, i = ii, \ldots, N$ should be satisfied;
for $j = ii : N$
if $B_j \leq \varepsilon$ ($\varepsilon$ is an arbitrarily small value) or $j \geq N - 1$
for $i = ii : j$
$p_i, s_i, r_i$ are decided by $p_{s,k}$ and $p_{r,k}$;
$B_i = B_{i-1} + \alpha s_i - r_i$;
$p_{si} = p_{s,k}$; $p_{ri} = p_{r,k}$;
end for
$ii = j + 1; k = k + 1$;
break;
end if
end for
end for
3) for $i = 1 : N - 1$
Compare the value of $p_o$ with $p_{si}$ and $p_{ri}$, then adjust $p_i, s_i, r_i, l_i$ and calculate $B_i$;
end for
4) $p_N = \max(p_o, (E_N + B_{N-1} - a)); l_N = \frac{E_N + B_{N-1}}{p_o + p_N}$.

---

## 5. Optimal Offline Power Allocation for Time-Varying Channel

Now we consider the time-varying channel scenario, and the instantaneous transmission rate is still (4), so the optimization problem P3 is

$$\max_{\{s_i, r_i, l_i\}} \frac{1}{N} \sum_{i=1}^{N} l_i R\left(\frac{E_i - s_i + r_i - al_i}{l_i}, h_i\right) \tag{33}$$

$$\text{s.t.} \sum_{j=1}^{i} (\alpha s_j - r_j) \geq 0, \ \ i = 1, \dots, N \tag{34}$$

$$E_i - s_i + r_i - al_i \geq 0, \ \ i = 1, \dots, N \tag{35}$$

$$s_i \geq 0, r_i \geq 0, 0 \leq l_i \leq 1, \ \ i = 1, \dots, N \tag{36}$$

P3 is still a convex optimization problem, and so we can also use the Lagrange multiplier method and the KKT conditions:

$$\frac{\partial L}{\partial s_i} = -\frac{h_i}{1 + h_i p_i{}^*} + \alpha \sum_{j=i}^{N} \lambda_j - \mu_i + \nu_i = 0, \ \ i = 1, \dots, N \tag{37}$$

$$\frac{\partial L}{\partial r_i} = \frac{h_i}{1 + h_i p_i{}^*} - \sum_{j=i}^{N} \lambda_j + \mu_i + {}_i = 0, \ \ i = 1, \dots, N \tag{38}$$

$$\frac{1}{2} \log(1 + h_i p_i{}^*) = \frac{h_i(p_i{}^* + a)}{2(1 + h_i p_i{}^*)} + a\mu_i - \theta_i + \sigma_i, \ \ i = 1, \dots, N \tag{39}$$

Other KKT conditions are the same as in (10), (11) and (12).

From (37) and (38), we can see that

$$p_i{}^* = \left[V - \frac{1}{h_i}\right]^+ \tag{40}$$

where $V$ is the water level and can be written as

$$V = \frac{1}{\alpha \sum_{j=i}^{N} \lambda_j + \nu_i} \tag{41}$$

For (41), we still use the definitions in [13]:

$$V_{si} = \frac{1}{\alpha \sum_{j=i}^{N} \lambda_j}, i = 1, \dots, N \tag{42}$$

$$V_{ri} = \frac{1}{\sum_{j=i}^{N} \lambda_j}, i = 1, \dots, N \tag{43}$$

In (42) and (43), $V_{si}$ and $V_{ri}$ are two water-level thresholds and the function of them is similar with those of $p_{si}$ and $p_{ri}$. Obviously,

$$V_{ri} = \alpha V_{si} \tag{44}$$

Like the time-invariant channel, the optimal solution should satisfy

$$p_i{}^* = \min(V_{si} - 1/h_i, \max(V_{ri} - 1/h_i, [E_i/l_i{}^* - a]^+)) \tag{45}$$

$$s_i{}^* = [E_i/l_i{}^* - a - p_i{}^*]^+, r_i{}^* = [p_i{}^* - (E_i/l_i{}^* - a)]^+ \tag{46}$$

Unlike the time-invariant channel, the channel may be in a very bad situation in some time slots where $\frac{1}{h_i} > V_{si}$. If this happens, the power allocation should be

$$p_i{}^* = 0, s_i{}^* = E_i, r_i{}^* = 0 \tag{47}$$

The analysis of (39) is similar to the previous one, but the difference is that the value of $p_o$ changes with $h_i$, and so $p_i{}^*$ and $l_i{}^*$ should satisfy

$$p_i{}^* = \max(p_o(h_i), E_i - s_i{}^* + r_i{}^* - a) \tag{48}$$

$$l_i{}^* = \frac{E_i - s_i{}^* + r_i{}^*}{p_i{}^* + a} \tag{49}$$

In addition, the policy in the last time slot should be

$$p_N{}^* = \max(p_o(h_N), E_N + B_{N-1} - a) \tag{50}$$

$$l_N{}^* = \frac{E_N + B_{N-1}}{p_N{}^* + a} \tag{51}$$

According to the above analyses, the optimal power allocation algorithm also has a three-threshold structure and depends on the values of $V_{si}$, $V_{ri}$, $h_i$ and $p_o(h_i)$. Algorithm 2 is as follows:

---

**Algorithm 2.** Optimal solution for the time-varying channel

---

1) Calculate $p_o(h_i), i = 1, \ldots, N$, initialize:
$p_i = 0, r_i = 0, s_i = 0, l_i = 1, B_i = 0, i = 1, \ldots, N, ii = 1, k = 1$;
2) for $i = ii{:}1{:}N$ do
Linear search the largest $V_{s,k}$ and $V_{r,k}$ which can make $p_i, s_i, r_i, i = ii, \ldots, N$ feasible in (45), (46) or in (47) and
the condition $B_i \geq 0, i = ii, \ldots, N$ should be satisfied;
for $j = ii : N$
if $B_j \leq \varepsilon$ ($\varepsilon$ is an arbitrarily small value) or $j \geq N - 1$
for $i = ii : j$
$p_i, s_i, r_i$ are decided by $V_{s,k}$ and $V_{r,k}$;
$B_i = B_{i-1} + \alpha s_i - r_i$;
$V_{si} = V_{s,k}; V_{ri} = V_{r,k}$;
end for
$ii = j + 1; k = k + 1$;
break;
end if
end for
end for
3) for $i = 1 : N - 1$
$p_i = \max(p_o(h_i), p_i)$;
$l_i = \frac{E_i - s_i + r_i}{p_i + a}$;
end for
4) $p_N = \max(p_o(h_N), (E_N + B_{N-1} - a)); l_N = \frac{E_N + B_{N-1}}{p_N + a}$.

---

## 6. Online Power Allocation Policies

Inspired by the offline optimal algorithm, the online algorithm should have similar structural features. The offline optimal solution can calculate $p_s$, $p_r$, $V_s$ and $V_r$ exactly by the full information about the energy arrivals and channel states; then, the optimal power allocation is decided by these thresholds and $p_o$ or $p_o(h_i)$. However, when considering the online policies, we only know the

information of the present moment and past moment, and so these thresholds need to be predicted. The prediction equations in [13] are

$$\alpha \int_{p_s}^{+\infty} (e - p_s) f_E(e) de = \int_0^{p_r} (p_r - e) f_E(e) de \tag{52}$$

$$\int \int_0^{+\infty} \alpha [e - [V_s - \frac{1}{h}]^+] - [V_r - \frac{1}{h} - e]^+ f_{E,H}(e, h) dedh = 0 \tag{53}$$

In (52), $f_E(e)$ is the probability density function of $e$. In (53), $f_H(h)$ is the probability density function of $h$, $f_{E,H}(e, h)$ is the joint probability density function of $e$ and $h$, and $f_{E,H}(e, H) = f_E(e) \cdot f_H(h)$ because the two variables are independent of each other.

The disadvantage of this prediction method is that the calculated thresholds are fixed, which is not feasible for all time slots and $\alpha$. Based on the solution of Equations (52) and (53), an adaptive-threshold algorithm is proposed in this paper. For the time-invariant channel, we first get $p_s$ from (52); then, the adaptive threshold should be

$$p_{si} = p_s(1 + \frac{(1 - \alpha)}{N - i + 1}) \tag{54}$$

In (54), two factors will affect the threshold: the storage efficiency $\alpha$ and time slot $i$. Obviously, the value of the threshold will increase when $\alpha$ becomes smaller or $i$ becomes larger, and the strategy is reasonable because the transmitter should use more energy instead of storage when a time slot is close to the end or the storage efficiency $\alpha$ is small. In addition, $p_{ri}$ is calculated by (22), according to the analysis in the offline part; then, the online power allocation policy should be

$$p_i = \begin{cases} \max(p_o, p_{si}), & E_i - a > p_{si} \\ \max(p_o, E_i - a), & p_{ri} \leq E_i - a \leq p_{si} \\ \max(p_o, E_i + \min(B_{i-1}, r_i) - a), & E_i < p_{ri} \end{cases} \tag{55}$$

$$l_i = \frac{E_i - s_i + \min(r_i, B_{i-1})}{p_i + a} \tag{56}$$

For the time-varying channel, we first get $V_s$ and then

$$V_{si} = V_s(1 + \frac{(1 - \alpha)}{N - i + 1}) \tag{57}$$

$V_{ri}$ is calculated by (44) and the online power allocation method is

$$p_i = \begin{cases} 0, & \frac{1}{h_i} > V_{si} \\ \max(p_o(h_i), V_{si} - \frac{1}{h_i}), & E_i - a + \frac{1}{h_i} > V_{si}, \frac{1}{h_i} \leq V_{si} \\ \max(p_o(h_i), E_i - a), & V_{ri} \leq E_i - a + \frac{1}{h_i} \leq V_{si} \\ \max(p_o(h_i), E_i + \min(B_i, r_i) - a), & E_i - a + \frac{1}{h_i} < p_{ri} \end{cases} \tag{58}$$

$$l_i = \frac{E_i - s_i + \min(r_i, B_{i-1})}{p_i + a} \tag{59}$$

## 7. Simulation Results and Analysis

The simulation tool is MATLAB 2014a in this paper (part of our MATLAB code is available in the Supplementary File). The system model is as shown in Figure 1 in part 3. The additive white Gaussian noise (AWGN) channel is adopted and we assume that the noise spectral density $N_0 = 10^{-19}$W/Hz, the bandwidth is 1 MHz and the path loss $h$ between the transmitter and the receiver is -100 dB for the time-invariant channel. For the time-varying channel, the $h_i$ obeys exponential distribution with $E[h_i] = -100$ dB. The energy arrival of each time slot is independent and obeys the uniform

distribution. We consider 10 time slots, and the length of each of time slot is 1 second. $B_0 = 0$ and the capacity of the battery has no limit. The circuit consumption power is set to be 5 mW.

Figure 2 shows the performance changes of each algorithm as the storage efficiency of the battery is changed when the channel is time-invariant. The energy arrivals obey the uniform distribution of [5, 15] mJ. The non-storage strategy means that the energy of the current slot is completely expended in the slot and the power allocation policy is as in (27) and (28). It can be seen from the figure that the algorithm proposed in this paper is obviously superior to the algorithm in [13] because compared with the algorithm in [13], the factor of circuit consumption is considered in our optimization process. The curve of the adaptive-threshold online algorithm proposed in this paper is close to the offline optimal strategy curve while the gap between the double-threshold offline algorithm in [13] and its corresponding fixed-threshold online algorithm is larger, which shows that the adaptive-threshold is more beneficial to improve the throughput. From Figure 2, it can be also found that under the low storage efficiency, the strategy of non-storage has a great advantage, and with the increasing of storage efficiency, the disadvantage of non-storage strategy is gradually reflected.

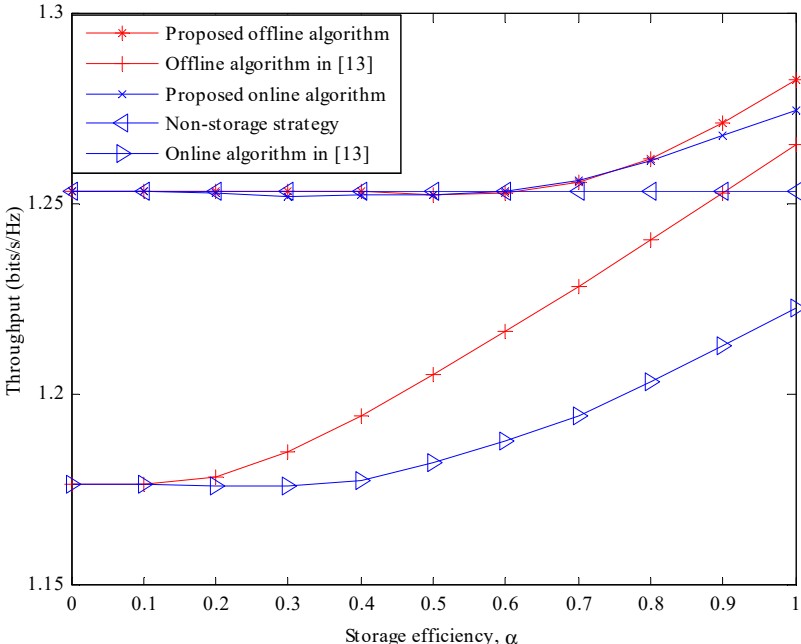

**Figure 2.** Throughput for the time-invariant channel with energy arrivals obeying the uniform distribution of [5, 15] mJ.

Figure 3 shows the performance of each algorithm when the energy arrivals obey the uniform distribution of [5, 20] mJ and the channel is time-invariant. It can be found that the overall change trend of each algorithm is similar to Figure 2. However, because of the increasing average value of energy arrivals, the overall throughput is obviously improved and the gap between the various algorithms gets larger. It is worth noting that with the improvement of storage efficiency, the double-threshold algorithm curve in [13] is closer to the curve of the offline algorithm proposed in this paper. This is due to the improvements of the mean of the energy arrivals and the storage efficiency; the energy used in the transmission increases, while the circuit consumption is constant, which makes the influence of circuit consumption smaller.

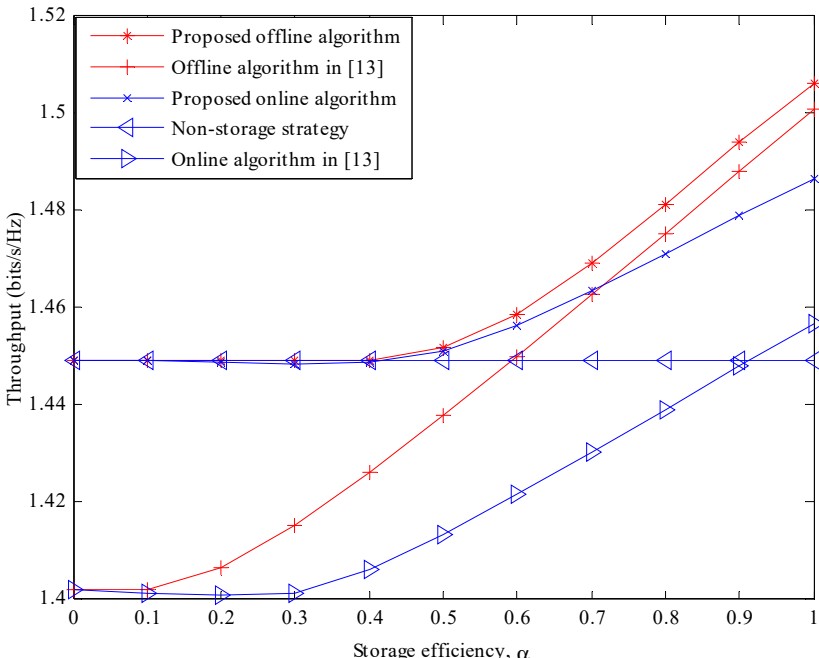

**Figure 3.** Throughput for the time-invariant channel with energy arrivals obeying the uniform distribution of [5, 20] mJ.

As shown in Figures 4 and 5, the performance of each algorithm varies with the change of storage efficiency under the time-varying channel when the energy arrivals obey the uniform distribution of [5, 15] mJ and [5, 20] mJ, respectively. As can be seen from the pictures, the algorithm proposed in this paper is still superior to the algorithm in [13]. However, the performance of the strategy of non-storage is not ideal, because the channel state is changing all the time and the strategy of non-storage may still transmit data when the channel is in a very poor state. As with the time-invariant channel scene, the adaptive-threshold online algorithm curve proposed in this paper is close to the curve of the offline algorithm. When the mean of energy arrival gets larger and the storage efficiency increases, the influence of circuit consumption decreases, and so the performance of the double-threshold algorithm in [13] will be closer to the algorithm in this paper.

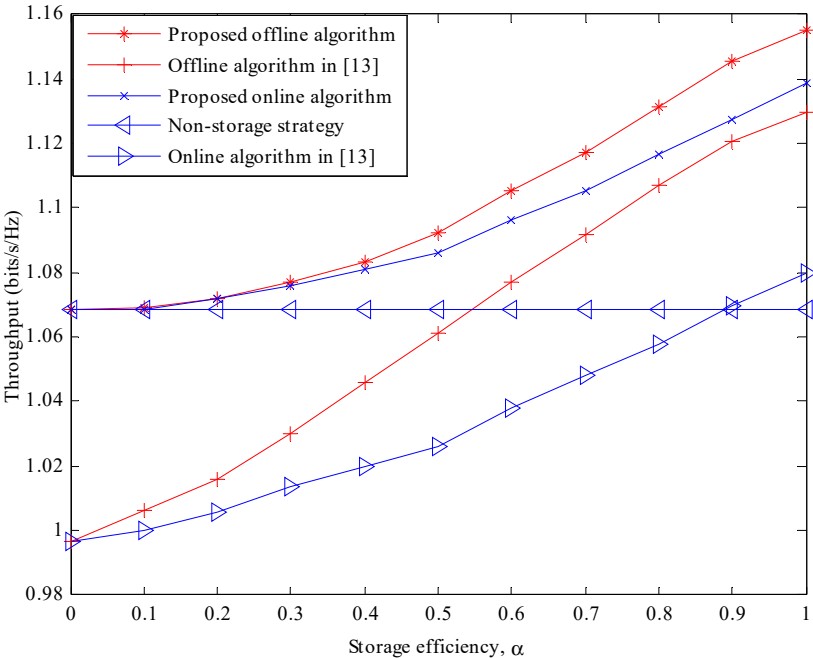

**Figure 4.** Throughput for the time-varying channel with energy arrivals obeying the uniform distribution of [5, 15] mJ.

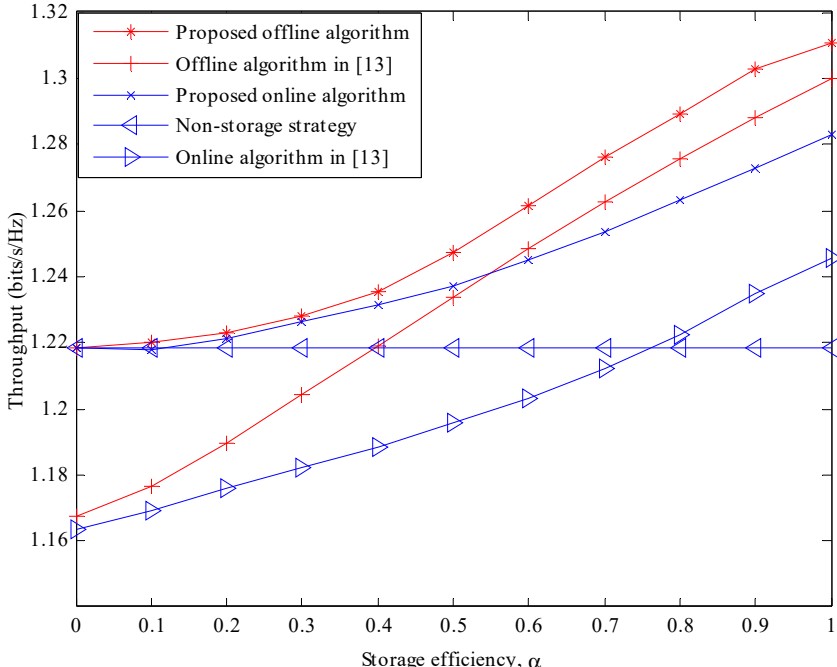

**Figure 5.** Throughput for the time-varying channel with energy arrivals obeying the uniform distribution of [5, 20] mJ.

By changing the *h*, we get three curves in Figure 6. It can be seen that the channel state plays a vital role and the performance of the proposed offline algorithm increases more obviously when the channel state improves. Figure 7 shows the performance of each algorithm when the storage efficiency is 60% and the channel is time-invariant. The performance of the online algorithm in [13] is the worst, while the proposed offline algorithm in this paper is still the best.

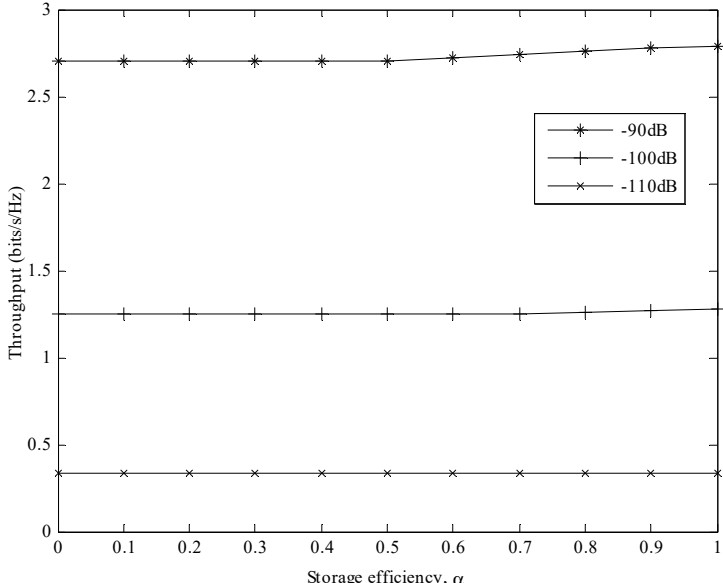

**Figure 6.** Throughput of the proposed offline algorithm for the time-invariant channel with energy arrivals obeying the uniform distribution of [5, 15] mJ.

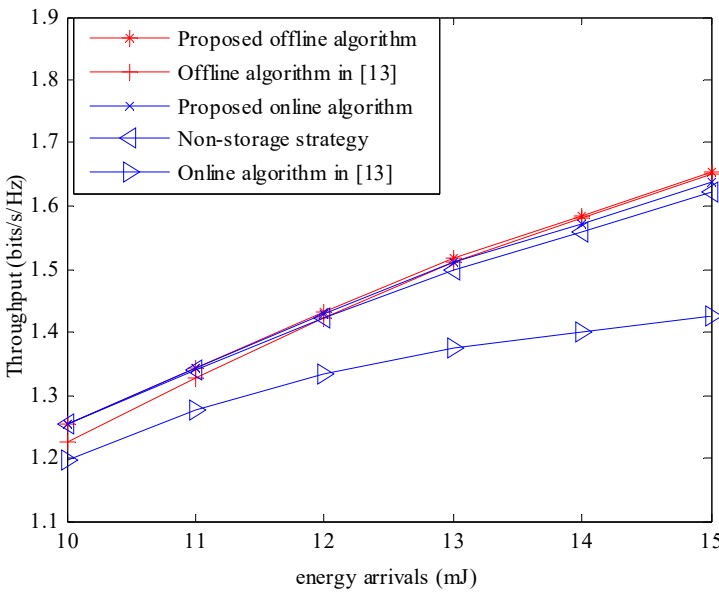

**Figure 7.** Throughput for the time-invariant channel with 60% storage efficiency.

## 8. Discussion

In this paper, convex optimization approaches such as the Lagrange multiplier method and the KKT (Karush–Kuhn–Tucker) condition are used to solve the problem we proposed. Then, we discuss and analyze the form of the optimal solution and find that the power allocation depends on three thresholds. Two of the thresholds can be calculated by a liner search while the third is decided by a channel statement. We next summarize the offline algorithms for both time-invariant and time-varying channels. Furthermore, inspired by offline algorithms, the online algorithm is proposed. The offline algorithm can adaptively adjust thresholds when the time and channel statements have changes. In addition, not only does the online algorithm have low complexity, but also the performance of the online algorithm is close to that of the offline one. Due to limited time and energy, we assume that the capacity of the battery is infinite and the maximum transmit power has no limit which will affect the optimization. These factors will be taken into account in future research work. The adaptive-thresholds

formula (54) and (57) are only applied to short-time transmission, and we will research a more universal online algorithm in future.

## 9. Conclusions

In this paper, aiming to maximize the average throughput of the system within a certain time, we propose the power allocation policies for the data transmission in a wireless communication system with energy consumption and storage losses at the same time. Firstly, the system model is set up; then, the optimization questions are proposed and the Lagrange multiplier method and KKT conditions are used to solve the problems. Based on the analysis of the expression of the optimal solution, the offline algorithm with a three-threshold structure is obtained. Furthermore, the online power allocation policy is studied and an adaptive-threshold online algorithm is proposed. Finally, the simulation results show that the proposed offline algorithm is better than other types of algorithms, and the online algorithm proposed is close to the optimal offline algorithm. By using these algorithms, we can significantly improve the system throughput and increase energy usage.

**Supplementary Materials:** The following are available online at http://www.mdpi.com/1999-4893/12/1/25/s1. Part of the MATLAB code is available in the Supplementary File).

**Author Contributions:** The manuscript was written by G.H. under the guidance of S.Z. and Q.Z. All authors reviewed the manuscript.

**Acknowledgments:** This work is supported by National Natural Science Foundation of China (61571234, 61631020) and the open research fund of National Mobile Communications Research Laboratory, Southeast University (No.2015D10).

**Conflicts of Interest:** The authors declare no conflict of interest.

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
