# Peer review of "Power Allocation Algorithm for an Energy-Harvesting Wireless Transmission System Considering Energy Losses"

_algorithms, doi:10.3390/a12010025_

Round 1

Reviewer 1 Report

The paper is almost identical with the one uploaded by authors approx. 2 months ago. It is focused on the proposal of two algorithms. The authors are analyzing the optimal solution for the time-invariant channel and for the time-variant channel within the topic of power allocation for EH wireless transmission systems.

There are slight differences in the introduction and a new paragraph (7 lines) is added to the chapter 7 – Simulation results and analysis (8 lines) and chapter 8 – Discussions (5 lines). Additional item are also in the references.

A chapter with experimental results is still missing. Due to my opinion, the experimental verification at least of a base idea of the simulation is more than essential for the technical paper.

Author Response

We are very grateful for your comments on the manuscript.

(1) We are sorry about that. Due to limited conditions, we really cannot do experimental verification in laboratory environment. The algorithm in wireless communication system is hard to verify. People who research in this filed only do simulations with software like MATLAB to verify. Some classic reference which are similar to our paper in this field also only do simulations:

1.    Jing, Y.; Ulukus, S. Optimal Packet Scheduling in an Energy Harvesting Communication System. IEEE Transactions on Communications 2012, 60, 220-230.

2.    Tutuncuoglu, K.; Yener, A. Optimum Transmission Policies for Battery Limited Energy Harvesting Nodes. IEEE Transactions on Wireless Communications 2012, 11, 1180-1189.

3.    Sharma, V.; Mukherji, U.; Joseph, V.; et al. Optimal Energy Management Policies for Energy Harvesting Sensor Nodes. IEEE Transactions on Wireless Communications 2008, 9, 1326-1336.

4.    Ozel, O.; Tutuncuoglu, K.; Yang, J.; et al. Transmission with Energy Harvesting Nodes in Fading Wireless Channels: Optimal Policies. IEEE Journal on Selected Areas in Communications 2011, 29, 1732-1743.

5.    Ho, C.K; Zhang, R. Optimal Energy Allocation for Wireless Communications With Energy Harvesting Constraints. IEEE Transactions on Signal Processing 2012, 60, 4808-4818.

6.    He, P.; Zhao, L.; Zhou, S.; et al. Recursive Waterfilling for Wireless Links With Energy Harvesting Transmitters. IEEE Transactions on Vehicular Technology 2014, 63, 1232-1241.

7.    Devillers, B.; Gündüz, D. A general framework for the optimization of energy harvesting communication systems with battery imperfections. Journal of Communications & Networks 2012, 14, 130-139.

8.    Xu, J.; Zhang, R. Throughput Optimal Policies for Energy Harvesting Wireless Transmitters with Non-Ideal Circuit Power. IEEE Journal on Selected Areas in Communications 2014, 32, 322-332.

9.    Wang, X.; Nan, Z.; Chen, T. Optimal MIMO Broadcasting for Energy Harvesting Transmitter With non-Ideal Circuit Power Consumption. IEEE Transactions on Wireless Communications 2015, 14, 2500-2512.

10.  Orhan, O.; Gunduz, D.; Erkip, E. Energy Harvesting Broadband Communication Systems With Processing Energy Cost. Wireless Communications IEEE Transactions 2013, 13, 6095-6107.

11.  Tutuncuoglu, K.; Yener, A.; Ulukus, S. Optimum Policies for an Energy Harvesting Transmitter Under Energy Storage Losses. IEEE Journal on Selected Areas in Communications 2012, 33, 467-481.

12.  Hsu, J. Power management in energy harvesting sensor networks. Acm Transactions on Embedded Computing Systems 2007, 6, 32.

Reviewer 2 Report

The introduction and research background is too short. The authors are suggested to improve the section Introduction with more focus on research background and motivation.

In the simulation, the authors are suggested to explain why the simulation results are observed.

The section Discussion is too short, looks like unnecessary. If the authors wanted to add something to discuss, I would suggest to focus on the design features, constraints, and potential improvement in more details.

Author Response

We are very grateful to your comments for the manuscript, we have rewritted discussion section:

In this paper, convex optimization knowledge such as the Lagrange multiplier method and the KKT (Karush-Kuhn-Tucker) condition are used to solve the problem we proposed. Then we discuss and analyze the form of optimal solution and find that the power allocation depends on three thresholds. Two of the thresholds can be calculated by liner search while the third is decided by channel statement. We next summarize the offline algorithms for both time-invariant and time-varying channel. Furthermore, inspired by offline algorithms, the online algorithm is proposed. The offline algorithm can adaptively adjust thresholds when the time and channel statement have changes. In addition, not only the online algorithm has low complexity but also the performance of online algorithm is close to that of the offline one. Due to limited time and energy, we assume that the capacity of battery is infinite and the max transmit power has no limit which will affect the optimization. These factors will be taken into account in future research work. The adaptive-thresholds formula (48) and (51) are just applied to short time transmission and we will research more universal online algorithm in future.

Reviewer 3 Report

The paper is well written. The proposed algorithms are novel enough and are in par or better than state of the art algorithms. Authors take into account more potential energy losses than previous approaches. Therefore, I recommend to accept this paper.

Some remarks: the mathematical part is well-known, but you need to double-check the formulas for final version. For example the definitions of p_i in Formulas (2) and (5a) are different.

Author Response

Thank you for your advice. The formula (2) is the definition of p_i: while (5a) is the throughput of the system:   where  is the instantaneous transmission rate: .So the definitions of p_i are same.

Round 2

Reviewer 1 Report

Dear authors,

Despite your arguments, I believe that a technically oriented article focusing on your subject must have a practical or experimental basis.
However, I will ask the editorial office to prefer the opinion of the remaining reviewers.

Author Response

We are sorry about that. We provide part of our MATLAB code in attachment.